# Short Course of Antibiotic Therapy for Gram-Negative Bacilli Bacteremia in Patients with Cancer and Hematopoietic Stem Cell Transplantation: Less Is Possible

**DOI:** 10.3390/microorganisms11020511

**Published:** 2023-02-17

**Authors:** Fabián Herrera, Diego Torres, Alberto Carena, Federico Nicola, Andrés Rearte, Elena Temporiti, Laura Jorge, Ricardo Valentini, Florencia Bues, Silvia Relloso, Pablo Bonvehí

**Affiliations:** 1Infectious Diseases Section, Centro de Educación Médica e Investigaciones Clínicas (CEMIC), Buenos Aires C1431, Argentina; 2Microbiology Laboratory, Centro de Educación Médica e Investigaciones Clínicas (CEMIC), Buenos Aires C1431, Argentina; 3Internal Medicine Department, Centro de Educación Médica e Investigaciones Clínicas (CEMIC), Buenos Aires C1431, Argentina

**Keywords:** Gram-negative bacteremia, cancer, short antibiotic course

## Abstract

Data about short courses of antibiotic therapy for Gram-negative bacilli (GNB) bacteremia in immunosuppressed patients are limited. This is a prospective observational study performed on adult patients with cancer and hematopoietic stem cell transplant (HSCT) who developed GNB bacteremia and received appropriate empirical antibiotic therapy (EAT), had a clinical response within 7 days and survived 48 h after the end of therapy. They received antibiotic therapy in the range of 7–15 days and were divided into short course, with a median of 7 days (SC), or long course, with a median of 14 days (LC). Seventy-four patients were included (SC: 36 and LC: 38). No differences were observed in baseline characteristics or in the presence of neutropenia: 58.3% vs. 60.5% (*p* = 0.84). Clinical presentation and microbiological characteristics were similar in SC and LC, respectively: clinical source of bacteremia 72.2% vs. 76.3% (*p* = 0.68); shock 2.8% vs. 10.5% (*p* = 0.35) and multidrug-resistant GNB 27.8% vs. 21.1% (*p* = 0.50). Overall, mortality was 2.8% vs. 7.9% (*p* = 0.61), and bacteremia relapse was 2.8% vs. 0 (*p* = 0.30). The length of hospitalization since bacteremia was 7 days (interquartile range (IQR), 6–15) for SC and 12 days (IQR, 7–19) (*p* = 0.021) for LC. In the case of patients with cancer or HSCT and GNB bacteremia who receive appropriate EAT with clinical response, 7 days of antibiotic therapy might be adequate.

## 1. Introduction

Bacteremia remains the major infection complication in immunosuppressed patients with solid tumors (ST), hematologic malignancies (HM), and hematopoietic stem cell transplant (HSCT), leading to an increment in mortality, length of hospitalization, and health care costs [1,2,3]. The incidence depends on the type of cancer and HSCT and the presence of risk factors for developing bacteremia, such as neutropenia, cytotoxic chemotherapy, intravenous catheters, and mucosal damage, among others. A change in epidemiology has been observed in the last decade, with a predominance of Gram-negative bacilli (GNB) as the main cause of bacteremia, especially in neutropenic patients [4,5]. Thus, the worldwide emergence of multidrug-resistant GNB (MDR-GNB) has become one of the most challenging healthcare system threats, especially in immunosuppressed patients. In this sense, GNB bacteremia caused by extended-spectrum β-lactamases (ESBL) Enterobacterales is the most frequent, and carbapenem-resistant Enterobacterales (CRE) mainly mediated by the presence of a carbapenemase gene are the most severe infections in many regions of the world [6,7]. The genes encoding for carbapenemases are typically located on plasmids and can be transferred both within bacterial species and across different species and genera [8,9,10]. *Pseudomonas aeruginosa*, the third most common pathogen in neutropenic patients, has also become a significant clinical problem because of the high levels of intrinsic and acquired resistance to many antibiotics used [11,12].

The Argentine Group for the Study of Bacteremia in Cancer and Stem Cell Transplant (ROCAS) enrolled 1277 bacteremia episodes in patients with HM and HSCT, and 60% of the isolates were GNB (75% Enterobacterales). Resistance to meropenem between HM and HSCT was 18.4% vs. 26.4% [13]. In addition, of 332 episodes of bacteremia in patients with ST, 67% of the isolates were GNB (84% Enterobacterales), and 20% were MDR [14]. In view of this complex scenario, initiating an early and appropriate empirical antibiotic therapy (EAT) is not an easy task. Moreover, long prescription of a combination of broad-spectrum antibiotics is also controversial. This results in an increase in the rate of adverse effects, *Clostridioides difficile* and fungal infections, higher costs, longer length of hospital stay and emergence of MDR-GNB [15,16,17,18,19,20,21,22]. Since prolonged antibiotic therapy is not an evidence-based clinical practice, shortening antibiotic therapy courses is an important antibiotic stewardship strategy [23]. 

The ideal approach would be prescribing the shortest therapy able to provide a clinical cure without infection relapse and a lower risk of antibiotic resistance. In this regard, three randomized controlled trials demonstrated that in non-complicated GNB bacteremia, a 7-day vs. 14-day course of antibiotic therapy was non-inferior for clinical cure, infection relapse, and 30-day mortality [24,25,26]. However, randomized controlled trials comparing SC vs. LC antibiotic therapy for GNB bacteremia in immunosuppressed patients have not been published. Data supporting shorter antibiotic courses for GNB bacteremia in patients undergoing HSCT, antineoplastic chemotherapy, and neutropenia are scarce [27]. 

Our study was designed as part of a multicenter registry and was first carried out in 2014 in a prospective cohort of cancer and HSCT patients [28]. The aim was to compare epidemiological, clinical, and treatment characteristics, as well as the outcome of patients receiving SC vs. LC antibiotic therapy for GNB bacteremia. 

## 2. Materials and Methods

### 2.1. Setting, Patients, and Study Design

A prospective observational study performed in a university hospital in Buenos Aires, Argentina, specialized in the management of oncological and transplant patients. All the episodes of monomicrobial GNB bacteremia in adult patients (≥18 years of age) managed as inpatients from May 2014 to December 2019 were included, provided that the following criteria were met: patients presented (a) ST or HM treated with chemotherapy one month prior to admission, or biological agents six months prior to admission, or they had been receiving steroids (at a dose equal to or higher than prednisone 20 mg daily or equivalent, for at least two weeks prior to admission); or (b) allogeneic HSCT (with graft versus host disease at any time or without this disease in the first two years), or autologous HSCT (in the first year post-transplant). All the following criteria should also be met: (a) appropriate EAT; (b) clinical response within 7 days; and (c) survival 48 h after the end of therapy. They had received total antibiotic therapy in the range of 7–15 days and were divided into SC with a median of 7 days (IQR 7-7) or LC with a median of 14 days (IQR 14-14). Patients in the LC group were included at the beginning of the study, while those in the SC group were recruited in 2018 when shortening antibiotic therapy for bacteremia was implemented as part of our antimicrobial stewardship program, which is in accordance with ECIL-4 guidelines recommendation [29]. Patients with polymicrobial or recurrent bacteremia and those receiving palliative care were excluded from the analysis, as well as those with a source of bacteremia requiring prolonged treatment (endocarditis/endovascular infections, severe skin and soft-tissue infections, central nervous system infections and osteomyelitis), or with a clinical source that required surgery. 

Patients were identified by the Section of Infectious Diseases, which evaluates all those patients hospitalized with ST, HM, and HSCT who develop fever and/or signs and symptoms of infection. They were included in the study at the time of positive blood culture, whether they had started EAT or not, and were then prospectively followed on a daily basis by direct patient care. Data were obtained from electronic and paper medical records and direct patient care, with a double check made with microbiological records from the laboratory. Variables included patients’ characteristics, type of cancer and HSCT, stage of underlying disease, neutropenia, immunosuppressant drugs, previous and recent colonization with MDR-GNB, previous infection with MDR-GNB, type of previous antibiotic use, GNB isolates with their resistance mechanisms and resistance profile, clinical source of bacteremia, type of antibiotic prescribed as monotherapy and combined, mortality, recurrence of bacteremia and length of hospitalization after bacteremia. The EAT was prescribed according to the institutional guidelines designed by the Section of Infectious Diseases, which individualized therapy based on the local epidemiology, type of clinical source, the severity of clinical presentation, hemodynamic stability, and presence of risk factors for MDR-GNB. The EATs prescribed as monotherapy in both groups were mainly piperacillin-tazobactam and meropenem, while those prescribed as combined therapy were meropenem + colistin or amikacin. The Infectious Diseases Staff chose definitive therapy based on the GNB isolates and their antibiotic resistance profile. Patients were followed up by two designated investigators (by direct patient care in hospitalized cases, as outpatients, or by a phone call in cases of patients discharged) or until the patient’s death, provided that it occurred before (assessed by direct patient care in patients still hospitalized or by a local healthcare database). 

### 2.2. Definitions

Neutropenia was defined as an absolute neutrophil count < 500 cells/mm^3^. High-risk febrile neutropenia was defined according to clinical variables, and a Multinational Association for Supportive Care in Cancer (MASCC) score < 21 [30]. The clinical source of infection was determined based on the isolation of the bacteria in the suspected source and/or the associated clinical signs and symptoms. Recent antibiotic use was defined as any antibiotic administered 30 days before the episode of bacteremia for more than 48 h. Recent ICU admission was defined as an admission within 14 days prior to the episode of bacteremia and for at least 72 h. Colonization with multidrug-resistant GNB was defined as “previous” when it occurred within six months before hospitalization and “recent” when it was detected within one week of the episode of bacteremia. 

Bacteremia was classified as nosocomial, healthcare-associated, or community-acquired, according to Friedman et al. [31]. Recurrence of bacteremia was defined as a new episode of bacteremia with the same GNB and antibiotic-resistant profile identical to that observed within 30 days of treatment discontinuation. The EAT was adequate if one or more antibiotics were active in vitro against the isolated bacteria. In patients with ESBL-Enterobacterales, empirical therapy with piperacillin/tazobactam or cefepime alone was considered inadequate [32]. In any patient with GNB, empirical therapy with tigecycline as the only active drug was considered inadequate. Response to treatment on day 7 of therapy was defined as the absence of fever for at least 4 days, absence of hypotension, and clinical resolution of all signs and symptoms of infection. In catheter-related bacteremia, catheters were removed on the day of diagnosis. Mortality was related to infection provided that there was microbiological, histological, or clinical evidence of active infection.

### 2.3. Microbiological Studies

At least two samples of blood cultures were taken and inoculated in aerobic and anaerobic bottles (BD BACTEC™ Plus Aerobic/F and Plus Anaerobic/F) and monitored in the automatic system BD BACTEC (Becton Dickinson, Sparks, Maryland, USA) for a minimum incubation period of five days. Bacteremia was defined as the isolation of pathogenic bacteria in at least one bottle of blood culture. MDR-GNB was defined as a GNB resistant to three or more of the following categories of antibiotics: carbapenems, piperacillin/tazobactam, third and fourth-generation cephalosporins, aztreonam, fluoroquinolones, or aminoglycosides [33,34]. Microbiological identifications were made with MALDI-TOF (BD Bruker Microflex MALDI Biotyper, Bruker Daltonics, Bremen, Germany). Antibiotic susceptibility testing was performed by disk diffusion, epsilometric tests, and/or the BD Phoenix automated system (Becton Dickinson). Breakpoints and interpretation were according to the CLSI recommendations. In carbapenem-resistant bacteria, carbapenemase production was investigated by the Blue-Carba assay and/or the double disk synergy tests (with carbapenems disks placed close to a boronic acid disk for KPC and an EDTA disk for identification of metallo-β-lactamases). The presence of genes coding for major carbapenemases (i.e., *bla*_VIM_, *bla*_NDM_, *bla*_IMP_, *bla*_KPC,_ and *bla*_OXA-48-group_) was investigated by a multiplex polymerase chain reaction (PCR) using specific primers [35]. In order to detect colonization with carbapenemase-producing Enterobacterales, ESBL-producing Enterobacterales, and multidrug-resistant *Pseudomonas aeruginosa*, rectal swabs were routinely collected (once a week and in every pre-transplant evaluation) and seeded in appropriate chromogenic media (CHROMAgar, Paris, France). Additionally, a multiplex PCR was performed directly from rectal swabs in order to detect *bla*_KPC_ and *bla*_OXA-48-group_.

*Clostridioides difficile* was investigated in every patient with diarrhea by immunochromatography for the presence of glutamate dehydrogenase (GDH) antigen and toxins A and B using a commercial kit (*C. Diff* Quick Check Complete TECHLAB Inc, Blacksburg, Virginia, USA). Those samples with positive GDH and negative toxins were analyzed by real-time PCR (RT-PCR), using the LightMix^®^ Kit *Clostridium difficile* EC in the Light Cycler 2.0 equipment (LC, Roche Diagnostics, Mannheim, Germany) [36,37].

### 2.4. Statistical Analysis 

The study population was characterized by descriptive statistics. For continuous variables, centrality (median) and dispersion (IQR) measures were used according to the distribution of variables. Categorical variables were analyzed using absolute frequency and percentage. Groups were compared using the U Mann–Whitney test for continuous variables and the Fisher exact test or the chi-square test for categorical variables. For all tests, a 95% level of statistical significance was used. The analyses were performed with the SPSS (Statistics for Windows, Version 22.0. Armonk, NY, USA) software packages. 

## 3. Results

A total of 175 patients with GNB bacteremia were evaluated, and 101 were excluded because they failed to meet the eligibility criteria: 23 polymicrobial bacteremias, 16 inadequate EAT, 20 absence of a clinical response within 7 days, 7 treatment duration > 15 days, and 35 treatment length between 9 and 13 days. The total study population consisted of 74 patients: 36 (48.6%) had HM (with lymphoma, 41.5%, and acute leukemia, 34%, being the most frequent), 21 (28.4%) had ST, and 17 (23%) had undergone HSCT (46.1% allogeneic). The most frequent stage of underlying cancer was recently diagnosed (18, 24.3%), and 54 (59.5%) patients had neutropenia, 88.6% classified as high risk by their MASCC score, with a median duration of 13 days. A total of 69 patients (93.2%) received chemotherapy one month before bacteremia, while 21 (28.4%) and 20 (27%) were treated with steroids and biological agents, respectively. Baseline characteristics of bacteremia episodes treated with SC and LC antibiotic therapy are outlined in Table 1.

Regarding epidemiological characteristics, 33 (44.6%) patients had recent hospitalization (1 month prior to bacteremia), 44 (59.5%) had recent antibiotics use, piperacillin-tazobactam being the most frequent (39.2%), 21 (28.4%) had previous and 24 (32.4%) had recent colonization with MDR-GNB, and 8 (10.8%) had previous isolation of MDR-GNB. Forty-one (55.4%) bacteremias were classified as nosocomial infections, and the length of hospitalization until bacteremia was 1 day (IQR: 0–14). The epidemiological findings of bacteremia treated with SC and LC antibiotic therapy are highlighted in Table 2.

The most frequent isolates were *E. coli* (31, 41.9%), *Klebsiella* spp. (25, 33.8%), and *P. aeruginosa* (5, 6.8%). Even though GNB distribution between groups reached no statistical significance, *E. coli* isolates were slightly more frequent in the SC group. Eighteen (24.30%) microorganisms were MDR, with ESBL-producing Enterobacterales (9, 12.2%) and KPC-carbapenemase-producing Enterobacterales (KPC-CPE) (8, 10.8%) being the most frequent, with similar distribution between the groups. As to resistance profile, more than 20% of the isolates were resistant to piperacillin-tazobactam and cefepime, and more than 10% to meropenem in both groups, which are the three most frequent EAT prescribed to those patients. Resistance to amikacin, colistin, tigecycline, and fosfomycin was low. All the 5 KPC-CPE isolates in the SC group proved to be susceptible to ceftazidime-avibactam. The microbiological characteristics and resistance profiles of bacteremia episodes receiving SC and LC antibiotic therapy are described in Figure 1, Figure 2 and Figure 3.

Of all the bacteremias analyzed, 55 (74.3%) had a clinical source, abdominal (colitis in all cases) (25, 33.8%) and catheter-related (13, 17.6%) being the most frequent. The EAT was monotherapy in 43 (58.1%) cases, being piperacillin-tazobactam the most frequent (22, 51.2%). Only patients under SC antibiotic therapy were prescribed ceftazidime-avibactam, and all of them received definitive treatment as monotherapy. In 11 (52.18%) of the 22 neutropenic patients in the SC group, antibiotic therapy was discontinued before neutrophil count recovery. 

A total of 72 patients (97.3%) had a fever, 23 (31.1%) presented hypotension, and 5 (6.8%) developed shock. All variables usually associated with mortality were comparable, except the APACHE II score was higher in the SC group. Thirty-day mortality was 5.4%, in no case related to infection. *Clostridioides difficile* infection and recurrence of bacteremia occurred in 4.1% (3 patients in the LC group) and 1.4% of the cases (only 1 patient with protracted neutropenia in the SC group), and the length of hospitalization after bacteremia had a median of 10 (7–15) days but was significantly shorter in the SC group. Differences observed in treatment and outcomes between the two groups are highlighted in Table 3. 

## 4. Discussion

The study assessed the SC of antibiotics as a feasible therapy for GNB bacteremia in immunosuppressed patients with cancer or HSCT who received adequate EAT with clinical response. They were compared to those who received LC antibiotic therapy. The total cohort comprised a high proportion of patients with neutropenia and complicated bacteremia, with clinical source, hypotension, and even shock. Both groups had similar baseline, microbiological, clinical, and epidemiological characteristics. A low mortality rate was observed in both the SC and the LC group. Half of the neutropenic patients discontinued antibiotic therapy before neutrophil recovery. The SC group had a significantly lower length of hospitalization after bacteremia, which could potentially reduce healthcare costs. Finally, as receiving antibiotic therapy for more than 7 days is one of the risk factors for CRE bacteremia, SC of antibiotics could reduce the risk of the emergence of these microorganisms and probably other MDR-GNB [28]. 

Treatment duration for GNB bacteremia in severely immunosuppressed patients, largely neutropenic, is still a controversial issue. Several guidelines on antimicrobial therapy in oncohematological and neutropenic patients have been published. They recommend adapting antibiotic therapy to local epidemiology, hemodynamic stability, source of infection, and risk factors for MDR-GNB [38,39]. Since a short course of antibiotic therapy is effective and safe for neutropenic patients with fever of unknown origin, most guidelines recommend this strategy to patients with 72 h or more of intravenous antibiotic therapy who have been hemodynamically stable since presentation and have been afebrile for 48 h or more [40,41,42]. However, a recommendation regarding antibiotic therapy duration for microbiologically documented infections is not uniformly established [29,30,43]. These discrepancies are evidenced by the real-life data on antimicrobial practices in febrile neutropenia. EBMT Infectious Disease Working Party surveyed over 567 centers from Europe and Asia. The answer to the question: What is the duration of antibiotic therapy in patients with positive blood cultures? was: <7 days, 1.1%, 7–10 days, 28%, 11–14 days, 36.4%, 15–21 days, 6%, and until the end of neutropenia, 30.2% [44].

A meta-analysis including the three randomized controlled trials compared the effect of short versus long treatment duration on all-cause mortality in pre-specified sub-groups. There were no differences in 30-day mortality in immunosuppressed patients. However, they had uncomplicated bacteremia, mostly from urinary sources, and the patients were hemodynamically stable. Furthermore, neutropenic patients and CRE bacteremia were excluded [45].

Our population differs from that included in the previous studies since more than half of our patients were neutropenic, and the clinical source of their bacteremia was other than the urinary tract. In addition, we included hemodynamically unstable patients with CRE bacteremia, mainly KPC-CPE. 

Recently, a retrospective cohort study compared the efficacy of short (median 6 days, IQR, 6–7) vs. long (median 11 days, IQR, 9.5–14) antibiotic courses for bacteremia in acute myeloid leukemia patients with febrile neutropenia. A total of 104 bacteremia episodes were included among 71 patients, and 46% received a short antibiotic course. Among the total population, 65% of the bacteremias were primary, 51% were caused by GNB, mostly Enterobacterales (ESBL, 23%), and there were no CRE isolates. The other isolated pathogens were coagulase-negative Staphylococci, 21%, and Streptococci, 13%. In both groups, similarities were observed regarding comorbidities, treatment phase of leukemia, duration of neutropenia, distribution of bacteria, and source of infection. Relapsing bacteremia within 30 days of antibiotic discontinuation was observed in 7.6% of the episodes; 5 of them received short-course treatment. The mortality rate 30 days after antibiotic discontinuation was 7.7%; only one death was considered of infectious origin and received long-course treatment [46].

Our study has some similarities with the abovementioned since patients who received SC and LC treatment were comparable in terms of baseline characteristics, clinical findings, outcome, and relapse of bacteremia. Notwithstanding that, our cohort is entirely made up of GNB, with more than 70% of the bacteremia largely presenting a clinical source. Moreover, all the epidemiological variables, as well as the isolated microorganisms with their resistance profile and mechanisms, have been widely described. Patients were all stratified according to the variables usually correlating with a high risk of overall mortality, such as the Charlson comorbidity index score, PITT score, and APACHE II score. Thus, patients in the SC group had a median APACHE II score higher than 20 points but a low mortality rate. This suggests that when patients receive adequate EAT with clinical response, shortening antibiotic therapy might be safe even in cases with a high risk of mortality. Unlike the previous study, ours described all the antibiotic regimens administered to the patients. In this sense, only patients in the SC group were treated with ceftazidime-avibactam because this antibiotic has been available in our country since 2018. All patients in the SC group finished their treatment as monotherapy as part of the antibiotic stewardship program implemented in our hospital. Finally, not only did we evaluate the outcome but also the length of hospitalization after the bacteremia episode, which was significantly reduced in the SC therapy.

We are aware of several limitations of the present study. First, the population analyzed was heterogeneous in terms of cancer type. However, some studies showed that the overall mortality of bacteremia episodes between HM and ST is similar or even higher in the latter group [47,48]. Second, neutropenic and non-neutropenic patients were included. Even though the presence of neutropenia could determine a high mortality rate during bacteremia episodes, some studies have shown no differences in overall mortality between neutropenic and non-neutropenic cancer patients [49,50]. In addition, the type and stage of cancer were equally distributed between SC and LC therapy, as well as the high risk and duration of neutropenia. Third, since most bacteremia episodes were caused by Enterobacterales, we did not apply the final results to other GNBs as non-fermenting. Fourth, the patients received different empirical antibiotic treatments, which could induce a bias in the outcome. Nevertheless, all received appropriate EAT, with a similar proportion of combined therapy and monotherapy. Fifth, since the sample size was small and a low prevalence of mortality was observed, we were unable to draw definitive conclusions regarding this issue. Notwithstanding that, both groups were comparable concerning baseline, clinical, and microbiological variables, as well as predictors of outcome.

The strengths of our study rely on its prospective design and the enrollment of only GNB, including carbapenem-resistant GNB. In addition, a high proportion of the bacteremia episodes had a clinical source that proved to be other than the urinary tract in most cases. 

To conclude, this study showed that 7-day antibiotic therapy might be adequate for patients with cancer and HSCT who developed GNB bacteremia due to Enterobacterales and received adequate empirical antibiotic therapy with clinical response. Moreover, this strategy could have potential benefits, such as reducing hospitalization and healthcare costs and the emergence of MDR-GNB. Further larger prospective studies are needed to confirm these findings and define the efficacy and safety of SC antibiotic therapy in this population. 

## Figures and Tables

**Figure 1 microorganisms-11-00511-f001:**
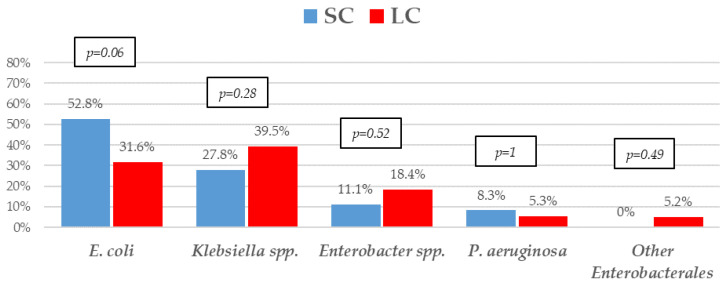
Etiology of Gram-negative bacilli bacteremia treated with a short course (SC) or long course (LC) of antibiotic therapy. *p*-values obtained by chi-square or Fisher’s exact test.

**Figure 2 microorganisms-11-00511-f002:**
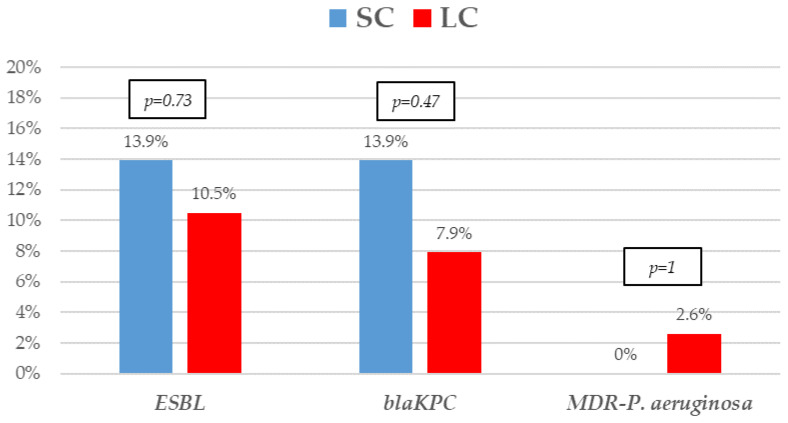
Resistance mechanisms of Gram-negative bacilli bacteremia treated with short-course (SC) or long-course (LC) antibiotic therapy. The proportion of extended-spectrum β-lactamases (ESBL), KPC carbapenem-producing (*bla*_KPC_), and multidrug-resistant *Pseudomonas aeruginosa. p*-values obtained by Fisher’s exact test.

**Figure 3 microorganisms-11-00511-f003:**
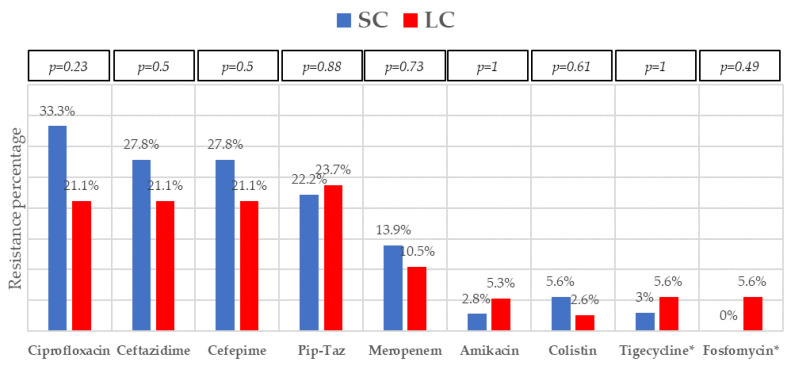
Resistance profile of Gram-negative bacilli bacteremia treated with short-course (SC) or long-course (LC) antibiotic therapy. Abbreviation: pip-taz, piperacillin-tazobactam. * Data obtained from Enterobacterales isolates (except *P. aeruginosa*). *p*-values obtained by chi-square or Fisher’s exact test.

**Table 1 microorganisms-11-00511-t001:** Baseline characteristics of patients with Gram-negative bacilli bacteremia.

Variables	SC *n* = 36 *n* (%)	LC *n* = 38 *n* (%)	** p*-Value
Age (years) (median, IQR)	57 (47–68)	60 (47–66)	0.71
Male sex	21 (58.3)	18 (47.4)	0.34
Inclusion criteria			
Hematologic malignancy	19 (52.8)	17 (44.7)	0.49
Solid malignancy	9 (25)	12 (31.6)	0.53
Hematopoietic stem cell transplant	8 (22.2)	9 (23.7)	0.88
Allogeneic HSCT	4 (50)	4 (44.4)	1
Type of hematologic malignancy			
Acute leukemia	7 (25.9)	11 (42.3)	0.21
Lymphoma	13 (48.2)	9 (34.6)	0.32
Myelodysplastic syndrome	3 (11.1)	4 (15.4)	1
Multiple myeloma	3 (11.1)	2 (7.7)	0.70
Chronic myeloproliferative neoplasm	1 (3.7)	0 (0)	1
Stage of underlying diseases			
Recently diagnosed	8 (22.2)	10 (26.3)	0.68
Complete remission	9 (25)	5 (13.1)	0.24
Partial remission	5 (13.9)	8 (21.1)	0.54
Refractory	6 (16.7)	8 (21.1)	0.63
Relapse	8 (22.2)	7 (18.4)	0.68
Recent chemotherapy (1 month prior to bacteremia)	35 (97.2)	34 (89.5)	0.18
Recent radiotherapy (1 month prior to bacteremia)	0 (0)	1 (2.6)	1
Current steroids use	11 (30.5)	10 (26.3)	0.68
Biological agents use	13 (36.1)	7 (18.4)	0.09
Charlson comorbidity index (median, IQR)	2 (2–4)	2 (2–3)	0.69
Neutropenia	21 (58.3)	23 (60.5)	0.85
High risk by their MASCC score	20 (95.2)	19 (82.6)	0.19
Neutropenia duration (days) (IQR)	14 (6–41)	12 (7–22)	0.59
Neutropenia > 10 days	14 (51.9)	14 (60.9)	0.52

Abbreviation: SC, short course; LC, long course; MASCC, Multinational Association for Supportive Care in Cancer; HSCT, hematopoietic stem cell transplant. * *p*-values obtained by chi-square or Fisher’s exact test for categorical variables and Mann–Whitney U-test for continuous variables.

**Table 2 microorganisms-11-00511-t002:** Epidemiological findings of Gram-negative bacteremia treated with SC or LC of antibiotic therapy.

Variables	SC *n* = 36 *n* (%)	LC *n* = 38 *n* (%)	** p*-Value
Recent hospitalization (1 month prior to bacteremia)	17 (47.2)	16 (42.1)	0.66
Previous colonization with KPC-CPE	5 (13.9)	1 (2.6)	0.10
Previous colonization with ESBL	7 (19.4)	4 (10.5)	0.34
Previous colonization with MDR-PA	1 (2.8)	3 (7.9)	0.61
Previous infection with MDR-GNB	3 (8.3)	5 (13.2)	0.71
Recent antibiotic use	20 (55.5)	24 (63.2)	0.51
Recent piperacillin-tazobactam use	13 (36.1)	16 (42.1)	0.56
Recent 3rd or 4th generation cephalosporin use	1 (2.8)	1 (2.6)	1
Recent carbapenem use	10 (27.8)	9 (23.7)	0.68
>7 days of antibiotic use prior to bacteremia	9 (25)	14 (36.8)	0.27
Fluoroquinolone prophylaxis	0 (0)	0 (0)	
Recent colonization with KPC-CPE	7 (19.4)	2 (5.3)	0.08
Recent colonization with ESBL	9 (25)	6 (15.8)	0.32
Recent colonization with MDR-PA	0 (0)	2 (5.3)	0.49
Recent intensive care unit admission	2 (5.6)	2 (5.3)	1
Central venous catheter in place	28 (77.8)	26 (68.4)	0.36
Urinary catheter in place	3 (8.3)	1 (2.6)	0.35
Nosocomial infection	19 (52.8)	22 (57.8)	0.66
Healthcare-associated infection	14 (38.9)	8 (21.1)	0.09
Community-associated infection	3 (8.3)	8 (21.1)	0.19
Length of hospitalization prior to bacteremia	0 (0–13)	5 (0–15)	0.27

Abbreviation: SC, short course; LC, long course; KPC-CPE, KPC-carbapenemase-producing Enterobacterales; ESBL, extended-spectrum β-lactamases; MDR-PA, multidrug-resistant *Pseudomonas aeruginosa*; MDR-GNB, multidrug-resistant Gram-negative bacilli. * *p*-values obtained by chi-square or Fisher’s exact test for categorical variables and Mann–Whitney U-test for continuous variables.

**Table 3 microorganisms-11-00511-t003:** Clinical presentation, treatment, and outcome of Gram-negative bacilli bacteremia treated with SC or LC of antibiotic therapy.

Variables	SC *n* = 36 *n* (%)	LC *n* = 38 *n* (%)	** p*-Value
Bacteremia with clinical source	26 (72.2)	29 (76.3)	0.69
Abdominal	14 (38.9)	11 (28.9)	0.36
Central venous catheter	6 (16.7)	7 (18.4)	0.84
Urinary tract	2 (5.6)	4 (10.5)	0.67
Respiratory tract	3 (8.3)	2 (5.3)	0.67
Skin and soft tissue	1 (2.8)	1 (2.6)	1
Perianal	1 (2.8)	3 (7.9)	0.61
Other	0	1 (2.6)	1
Fever	35 (97.2)	37 (97.4)	0.97
Hypotension	10 (27.8)	13 (34.2)	0.55
Empirical monotherapy antibiotic therapy	22 (61.1)	21 (55.3)	0.61
Piperacillin-tazobactam	10 (45.5)	12 (57.1)	0.44
Meropenem	9 (40.9)	7 (33.3)	0.61
Ceftriaxone	0 (0)	2 (9.5)	0.23
Ceftazidime-avibactam	2 (9.1)	0 (0)	0.49
Empirical combination antibiotic therapy	14 (38.9)	17 (44.7)	0.61
Meropenem + colistin	4 (28.6)	7 (41.2)	0.71
Meropenem + amikacin	2 (14.13)	6 (35.3)	0.23
Ceftazidime-avibactam + amikacin	6 (42.8)	0 (0)	0.004
Definitive monotherapy antibiotic therapy	36 (100)	33 (83.6)	0.024
Fluoroquinolones	5 (13.9)	11 (33.3)	0.08
Piperacillin-tazobactam	8 (22.2)	12 (36.4)	0.19
Meropenem	9 (25)	5 (15.2)	0.37
Ceftriaxone	6 (16.7)	5 (15.2)	1
Ceftazidime-avibactam	5 (13.9)	0 (0)	0.05
APACHE II score the day of bacteremia (median, IQR)	21 (19–23)	17 (14–20)	<0.001
PITT score the day of bacteremia (median, IQR)	1 (0–2)	1 (0–2)	0.22
Shock at presentation	1 (2.8)	4 (10.5)	0.36
30-day mortality	1 (2.8)	3 (7.9)	0.61
Infection-related mortality	0 (0)	0 (0)	
Recurrence of bacteremia	1 (2.8)	0 (0)	0.49
Length of hospitalization after bacteremia	7 (7–12)	12 (8–20)	0.02
*Clostridioides difficile* infection	0 (0)	3 (7.9)	0.24

Abbreviation: EAT, empirical antibiotic treatment; SC, short course; LC, long course. * *p*-values obtained by chi-square or Fisher’s exact test for categorical variables and Mann–Whitney U-test for continuous variables.

## Data Availability

Data is available upon request. Contact the corresponding author.

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
