# Peer review of "Short Course of Antibiotic Therapy for Gram-Negative Bacilli Bacteremia in Patients with Cancer and Hematopoietic Stem Cell Transplantation: Less Is Possible"

_microorganisms, 2023, doi:10.3390/microorganisms11020511_

Round 1

Reviewer 1 Report

Dear Editor,

Thank you for your peer review invitation. The manuscript, entitled “Short Course of Antibiotic Therapy for Gram-negative Bacilli Bacteremia in Patients with Cancer and Hematopoietic Stem Cell Transplantation: Less is Possible” is a prospective observational study performed on adult patients with cancer and hematopoietic stem cell transplant. In this study 74 patients were included and their baseline characteristics were similar. The authors found that for patients with cancer or HSCT and GNB bacteremia who receive appropriate EAT with clinical response, 7 days of antibiotic therapy may be enough. This is a meaningful study, however due to some drawbacks, my suggestion is major revision.

Comments:

1. When did the 74 patients who received a total antibiotics therapy in the range of 7-15 days participate this study? If they were surveyed at once after the antibiotics therapy? Did the authors follow up with the patients on an ongoing basis? Are the results of this study representative and indicative of the long-term condition of the patients?

2. In Figure 2, the p-value of blaKPC between SC and LC is 0.73. Could the authors introduce the method to calculate p-value in detail? Because it looks that the difference of blaKPC between two groups is significant. In addition, the percent of ESBL and blaKPC in SC is all 13.9%, the percent of ESBL and blaKPC in LC is 10.5% and 7.9%, respectively. The p-value of blaKPC should be smaller than the p-value of ESBL. I doubt the accuracy of the p-value in this study.

3. The background to the criteria for judging the health of patients is not sufficiently developed in the INTRODUCTION section. Please supplement some information on it. Also, the antibiotic resistance genes and their carriage in different species and strains. For example, representative gram negative strain Pseudomonas aeruginosa (doi: 10.1016/j.ebiom.2019.102599. DOI:10.1016/j.ijantimicag.2018.04.013. DOI: 10.1089/mdr.2020.0420. DOI:10.1016/j.ijantimicag.2017.09.008), Klebsiella pneumoniae (doi: 10.1093/jac/dkaa492, doi: 10.1080/22221751.2021.1984182.), and Enterobacter cloacae (DOI:10.1089/mdr.2017.0146. doi: 10.3389/fcimb.2020.00314.)

4. In results and discussion section, the description about figure 1, 2, and 3 is too less. And the authors did not do significant analysis in figure 1.

5. The blank line in table 1 should be deleted.

Reviewer 2 Report

This study is a prospective observational study on the duration of antibiotics for monobacterial Gram-negative bacilli bacteremia in immunocompromised hosts involving 74 patients (36 patients with short-course antibiotics, 38 patients with long-course antibiotics). This topic is of no doubt an important topic, as mentioned by the researchers that RCTs were available for immunocompetent hosts, but evidence for immunocompromised hosts is scarce. Overall this study is a good study given its prospective nature, and it is worth publishing. Here are some comments that are worth considering by the authors during revision.

Major comments:

1) Referring to Table 3, please double-check the data presented in this table. One patient is missing for bacteremia with a clinical source in the long-course antibiotics group.

2) It is observed that the majority of cases in this study were having bacteremia from the abdominal source. The authors may want to clarify what abdominal source means (i.e. whether it means surgical causes such as cholangitis/ cholecystitis or whether it includes medical causes such as colitis) and whether operative procedures were performed for those with surgical causes. This is important as we understand that patients, in general, require a shorter duration of antibiotics after definitive surgery (e.g. cholecystitis with cholecystectomy). 

3) Comment 2 also applies to bacteremia from the central venous catheters. The authors may want to clarify whether the central venous catheters were removed after identification of bacteremia, as a removed catheter requires a shorter duration of antibiotics when compared with a retained catheter, and it also affects the chance of relapse/ recurrence of bacteremia.

Minor Comments:

1) I agree with the limitation of your study that given the low number of Pseudomonas aeruginosa isolated from blood culture, the results may only apply to bacteremia due to Enterobacteriales but further research should be performed for non-glucose fermenters. This point should also be reflected in your conclusion statement +/- abstract.

2) Around 60% of your patients were neutropenic during the episodes of bacteremia. I am more interested in knowing how many percent of the patients were out of the neutropenic phase when the clinical team stopped the antibiotics, and whether there is any recurrence of infection or bacteremia after stopping antibiotics for patients still in the neutropenic phase.

3) "Cancer and hematopoietic stem cell transplantation" is a very heterogenous population, which includes patients with neutrophil defect (neutropenia due to chemotherapy), B cell defects (patients on immunotherapy such as Rituximab), and T cell defects (patients on high dose steroids and immunosuppressants). Given the heterogenous population, the sample size is actually relatively small to account for the heterogeneity of the issue. Having said that, given this study is prospective in nature, I still believe that the current study is important to provide preliminary evidence for large studies on this issue in the future.

Reviewer 3 Report

This Manuscript covers a very important topic and offers an insight into different practices in antimicrobial stewardship. I have a few comments and questions for the authors to consider.

Introduction is very well written and covers the topic quite well. I have one minor comment. Please define HSCT when mentioning it the first time in the Introduction section of the Manuscript as well as other abbreviations.

In the Methods section, were there any other exclusion criteria, could you provide the number of patients screened and excluded? Could you also write the antibacterials used in SC and LC group? This also may be important for the duration of treatment. Also, can follow-up be described in greater detail, i.e. proportion of patients followed via phone-call, was this done by a single investigator or?

Data presentation should be included in methods and tables footnotes.

Figure 1, is p-value missing?

The Discussion section may be reorganised to comment and discuss your results first. Considering a realtive small sample size, some findings of low p-values may also be commented on. 

Furthermore, are there any other conclusions that may be drawn from your results other than a shorther antibiotics course for clinicians?

Round 2

Reviewer 1 Report

Comments:

1. In this study seventy-four patients were observed (SC:36 and LC: 38). The number of samples is too less for an observational study. Do the authors think the results of this study are representative and the conclusion that 7 days of antibiotic therapy may be adequate is convincing?

2. In Figure 2, the p-value between two groups in MDR-P. aeruginosa is equal to 1. Please check the p-value in the text again and make sure they are all correct.

3. In Table 3, the data on OTHER, FEVER, and HYPOTENSION are not in line.

4. The format of the references is not consistent. For example, some references are attached with doi number, but some are not like reference 13. In addition, the format of doi number is inconsistent, like reference 14 and 28.

Author Response

Response to Reviewer 1 Comments second round.

We thank you for all your comments.

Point 1. In this study seventy-four patients were observed (SC:36 and LC: 38). The number of samples is too less for an observational study. Do the authors think the results of this study are representative and the conclusion that 7 days of antibiotic therapy may be adequate is convincing?

Response 1. We know that the sample size was small. However, since patients in the SC group had their underlying disease, were not in remission, were highly immunosuppressed, had an APACHE II score higher than 20, and those with neutropenia were mainly high-risk patients according to their MASCC score, the 30-day mortality rate of 2.8%, and in no case related to infection, is quite low. As mentioned in line 455, we are aware that this is one of the study limitations, so we cannot draw definitive conclusions in this regard. In addition, in the last statement in conclusions we mentioned “Further larger prospective studies are needed to confirm these findings and define the efficacy and safety of SC antibiotic therapy in this population”. Therefore, the statement “SC of therapy may be adequate” has been replaced by “SC of therapy might be adequate” in the abstract and conclusions.

Point 2. In Figure 2, the p-value between two groups in MDR-P. aeruginosa is equal to 1. Please check the p-value in the text again and make sure they are all correct.

Response 2. The p-value is correct. It was calculated with Fisher´s exact test.

Point 3. In Table 3, the data on OTHER, FEVER, and HYPOTENSION are not in line.

Response 3. It was corrected.

Point 4. The format of the references is not consistent. For example, some references are attached with doi number, but some are not like reference 13. In addition, the format of doi number is inconsistent, like reference 14 and 28.

Response 4. The references that do not have doi number have been presented in congresses (13,22,50). The doi number of the other references mentioned above has been modified (14,28,48).